# The influence of decision-making in tree ring-based climate reconstructions

Ulf Büntgen [1,2,3,4✉], Kathy Allen[5,6], Kevin J. Anchukaitis [7], Dominique Arseneault [8], Étienne Boucher[9,10,11], Achim Bräuning [12], Snigdhansu Chatterjee[13], Paolo Cherubini[2], Olga V. Churakova (Sidorova) [14], Christophe Corona[15,16], Fabio Gennaretti[17], Jussi Grießinger [12], Sebastian Guillet[16], Joel Guiot[18], Björn Gunnarson[19], Samuli Helama [20], Philipp Hochreuther[12], Malcolm K. Hughes [21], Peter Huybers[22], Alexander V. Kirdyanov [14,23], Paul J. Krusic[1,19], Josef Ludescher[24], Wolfgang J.-H. Meier[12], Vladimir S. Myglan[25], Kurt Nicolussi [26], Clive Oppenheimer [1,27], Frederick Reinig [28], Matthew W. Salzer[21], Kristina Seftigen [2,29], Alexander R. Stine [30], Markus Stoffel [16,31,32], Scott St. George [33], Ernesto Tejedor [34], Aleyda Trevino[22], Valerie Trouet[21], Jianglin Wang[35,36,37], Rob Wilson[38,39], Bao Yang[35,36,37], Guobao Xu[21,40] & Jan Esper[3,28]

Tree-ring chronologies underpin the majority of annually-resolved reconstructions of Common Era climate. However, they are derived using different datasets and techniques, the ramifications of which have hitherto been little explored. Here, we report the results of a double-blind experiment that yielded 15 Northern Hemisphere summer temperature reconstructions from a common network of regional tree-ring width datasets. Taken together as an ensemble, the Common Era reconstruction mean correlates with instrumental temperatures from 1794–2016 CE at 0.79 ($p < 0.001$), reveals summer cooling in the years following large volcanic eruptions, and exhibits strong warming since the 1980s. Differing in their mean, variance, amplitude, sensitivity, and persistence, the ensemble members demonstrate the influence of subjectivity in the reconstruction process. We therefore recommend the routine use of ensemble reconstruction approaches to provide a more consensual picture of past climate variability.

A full list of author affiliations appears at the end of the paper.

Absolutely dated tree-ring width (TRW) measurements from long-lived trees and relict wood (e.g., archaeological, remnant, historical and subfossil) are frequently used for the reconstruction of past climate variability[1–4]. The study of TRW variation in samples from cold high-elevation/-latitude sites can reveal changes in growing season temperature[1], while TRW chronologies from lower elevation, temperate and semiarid sites, where plant growth predominantly depends on soil moisture availability, more often represent hydroclimatic changes[4]. Yet, despite the broad geographic coverage and precise dating of extra-tropical tree-ring chronologies[5,6], there are only nine temperature-sensitive TRW chronologies in the Northern Hemisphere (NH) that span the past two millennia[3]; and far fewer in the Southern Hemisphere[2]. In addition to the paucity of multi-millennial TRW datasets from upper or northern treeline ecotones, subjectivity in site and series selection, correction for biological age trends in raw TRW measurements (hereafter referred to as detrending), and the climate calibration procedure, can have substantial consequences for the reconstruction of regional-scale to large-scale climate variability. The degree of biological memory in TRW chronologies may also affect the reconstruction's accuracy on interannual time scales[7,8]. This year-to-year bias is less pronounced in maximum latewood density (MXD) chronologies[9], but only one MXD-based summer temperature reconstruction—from northern Scandinavia —has so far been developed for the entire Common Era[10]. Moreover, the tree-ring literature would benefit greatly from more explicit, systematic and quantitative consideration of methodological and conceptual biases due to proxy properties and experimental techniques. This exercise is also considered only rarely in multi-proxy temperature reconstructions[11–15], whereas climate modellers have a long tradition of addressing uncertainties and stochastic processes in their simulations through ensemble approaches[14].

Here, we present a community-driven ensemble experiment to assess the influence of decision-making on the interannual variability and multi-centennial trajectory of climate reconstructions. Based on a double-blind approach that ensures conceptual and methodological independence between the participating laboratories, we show how different techniques of extracting climatic information from TRW data influence the final reconstructions.

## Results

**Decision making in temperature reconstructions**. Challenged with the same task to develop the most reliable NH summer temperature reconstruction for the Common Era from nine high-elevation/high-latitude TRW datasets[3] (Fig. 1), each of the 15 groups who contributed independently to this experiment (referred to here as R1–R15) have experience in developing tree ring-based climate reconstructions. However, each group employed a distinct reconstruction approach (Fig. 2; Supplementary Table 1), manifested in different series and site selections, detrending methods, temperature targets, and calibration techniques (see the "Methods" section).

There are many reasonable choices that investigators can make in developing climate reconstructions. Nine groups used all of the raw TRW measurements, whereas three groups removed particularly short series; one group identified and removed several duplicates, and another group reduced each dataset to 200 TRW series spanning the past 2000 years with equal annual sample replication. This last group also applied individual series detrending rather than any form of age-related composite detrending[16–18] (see the "Methods" section for detail).

After chronology development and signal detection at the site level, nine groups decided to combine the TRW chronologies from all nine sites in their final large-scale reconstructions, while five groups selected data from just seven sites, and one group from only five sites. The TRW data from NSC, ALP, YAM and ALT were used in all 15 reconstructions. QUL was used in 14 reconstructions, GTB in 13, TAI in 12, and data from SCO and NYA in only 11 reconstructions. Eleven groups identified mean June–August (JJA) temperatures as the optimal meteorological target season, whereas three groups selected 4-month averages as seasonal windows (i.e., May–September and June–October). One group chose the June–July temperature mean as optimal for

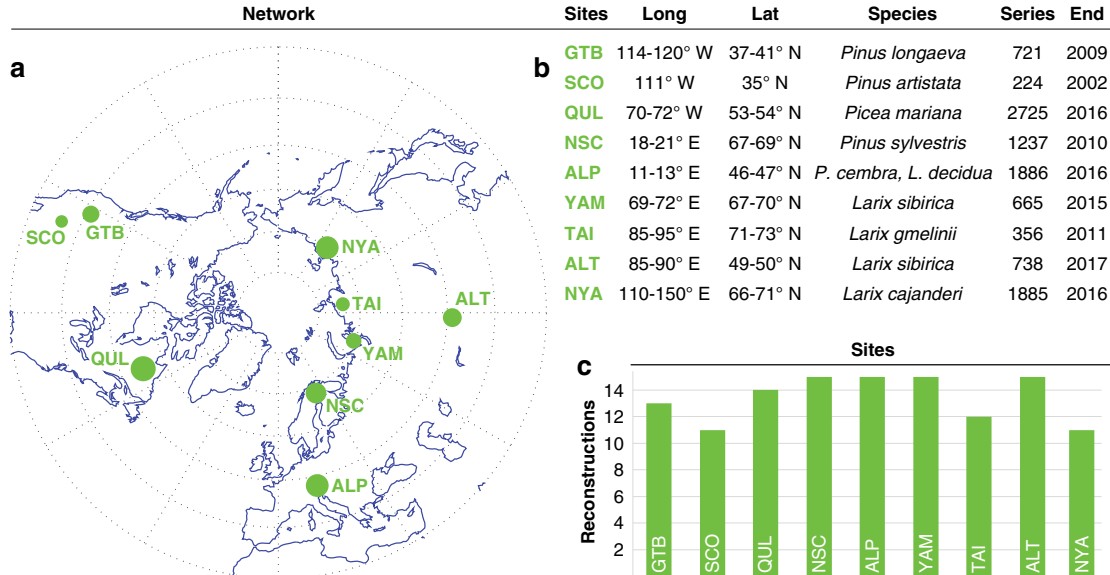

| Sites | Long | Lat | Species | Series | End |
|---|---|---|---|---|---|
| GTB | 114-120° W | 37-41° N | *Pinus longaeva* | 721 | 2009 |
| SCO | 111° W | 35° N | *Pinus artistata* | 224 | 2002 |
| QUL | 70-72° W | 53-54° N | *Picea mariana* | 2725 | 2016 |
| NSC | 18-21° E | 67-69° N | *Pinus sylvestris* | 1237 | 2010 |
| ALP | 11-13° E | 46-47° N | *P. cembra, L. decidua* | 1886 | 2016 |
| YAM | 69-72° E | 67-70° N | *Larix sibirica* | 665 | 2015 |
| TAI | 85-95° E | 71-73° N | *Larix gmelinii* | 356 | 2011 |
| ALT | 85-90° E | 49-50° N | *Larix sibirica* | 738 | 2017 |
| NYA | 110-150° E | 66-71° N | *Larix cajanderi* | 1885 | 2016 |

**Fig. 1 Dendro network . a** Spatial distribution of the 15 tree-ring width (TRW) sites with three of them in North America (GTB, SCO and QUL), two in Europe (NSC and ALP), three in northern Siberia (YAM, TAI and NYA), and one in inner Eurasia (ALT). Dot size represents TRW sample replication, ranging from 224 series in SCO to 2725 series in QUL. **b** Longitude and latitude, species, the total number of series, and last ring of the nine regional TRW datasets. **c** Consideration of the nine TRW datasets in the 15 ensemble reconstructions, ranging from 11 times (SCO and NYA) to 15 times (NSC, ALP, YAM, ALT).

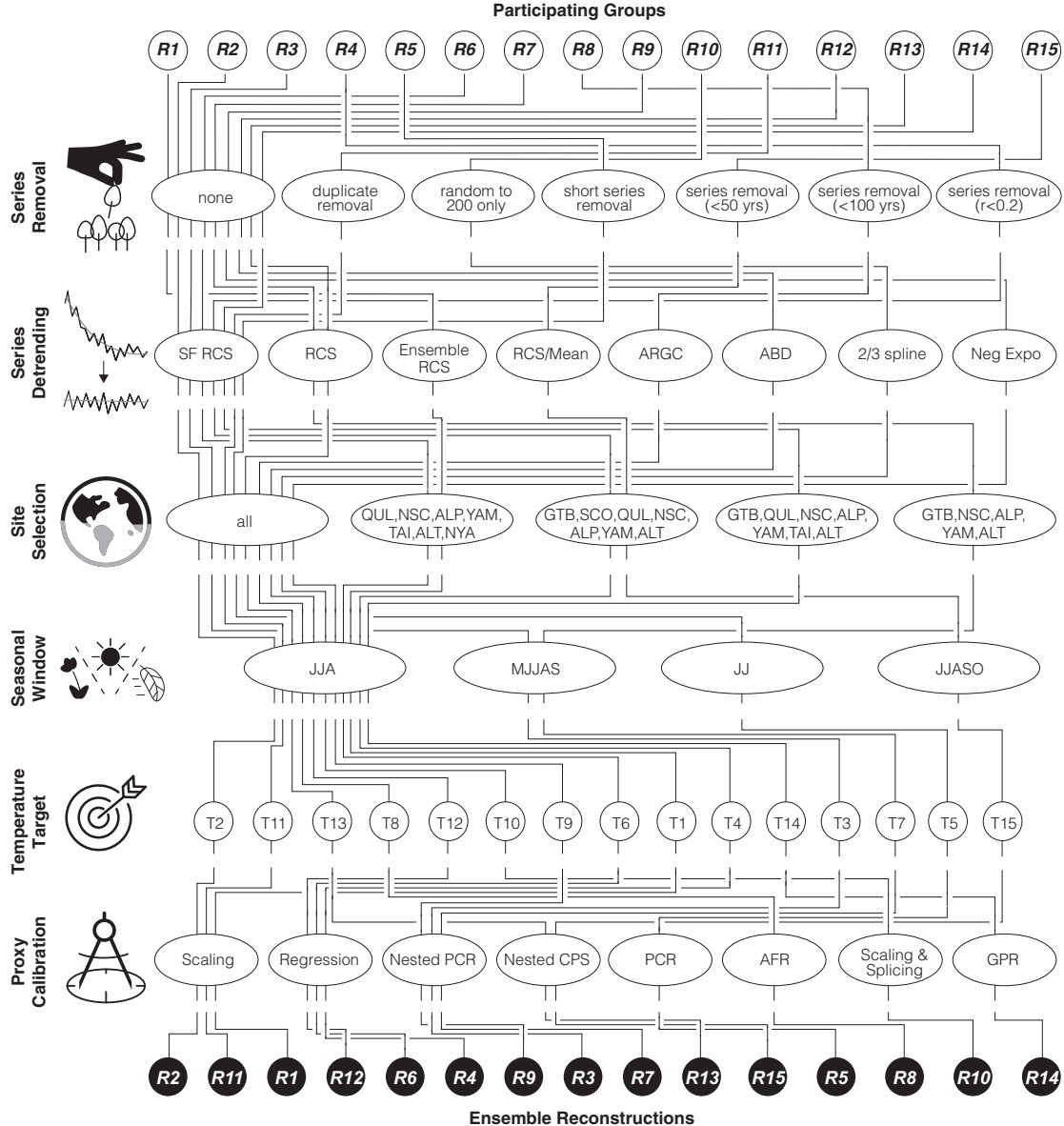

**Fig. 2 Ensemble approach.** Flow chart of the 15 large-scale Northern Hemisphere (NH) summer temperature reconstructions for the Common Era. SF RCS signal-free regional curve standardisation, ARGC adaptive regional growth curve, ABD age band decomposition for detrending, PCR principal component regressions, CPS composite plus scaling, AFR analogue frequency regression, and GPR gaussian process regression (see the "Methods" section and Supplementary Table 1 for details and further abbreviations).

calibration. No two temperature target datasets are the same. Each group identified different regions within the NH, and extracted different seasons between May and October within the years 1750 and 2016 from different gridded temperature products. All of these target data selection choices, however, reveal a statistically similar picture of large-scale summer temperature variability over the entire 20th and early 21st centuries (Fig. 3a). Different techniques of either scaling the TRW chronologies to the mean and variance of instrumental measurements or regressing the TRW chronologies against the instrumental measurements were applied (see Fig. 2 and Supplementary Table 1 for further details of the individual reconstruction methods).

**Differences between temperature reconstructions.** Despite substantial amplitude differences between the individual

ensemble reconstructions during two cold spells in the 1810s and 1830s (Fig. 3b), and during the recent warming since the 1980s, their mean and median track the instrumental measurements well between the end of the 19th century and circa 1990 CE (Fig. 3c). Proxy-target correlations are 0.76 and 0.79 for the reconstruction mean and median, respectively. Although the first-order auto-correlation (AC1) of the mean record of the 15 individual temperature targets is lower than that of the reconstruction mean (0.76 versus 0.88 from 1794–2016 CE), all split-period calibration/verification statistics of the reconstruction mean are strongly positive and temporally robust (Supplementary Table 2). Pearson's correlation coefficients for the early and late calibration/verification periods (1794–1905 and 1906–2015 CE) range from 0.67 to 0.75, while reduction of error (RE) and coefficient of efficiency (CE) statistics of the same split periods range from 0.44–0.74 and from 0.23–0.45, respectively. When using first-differences of the time-series, correlation coefficients decline to

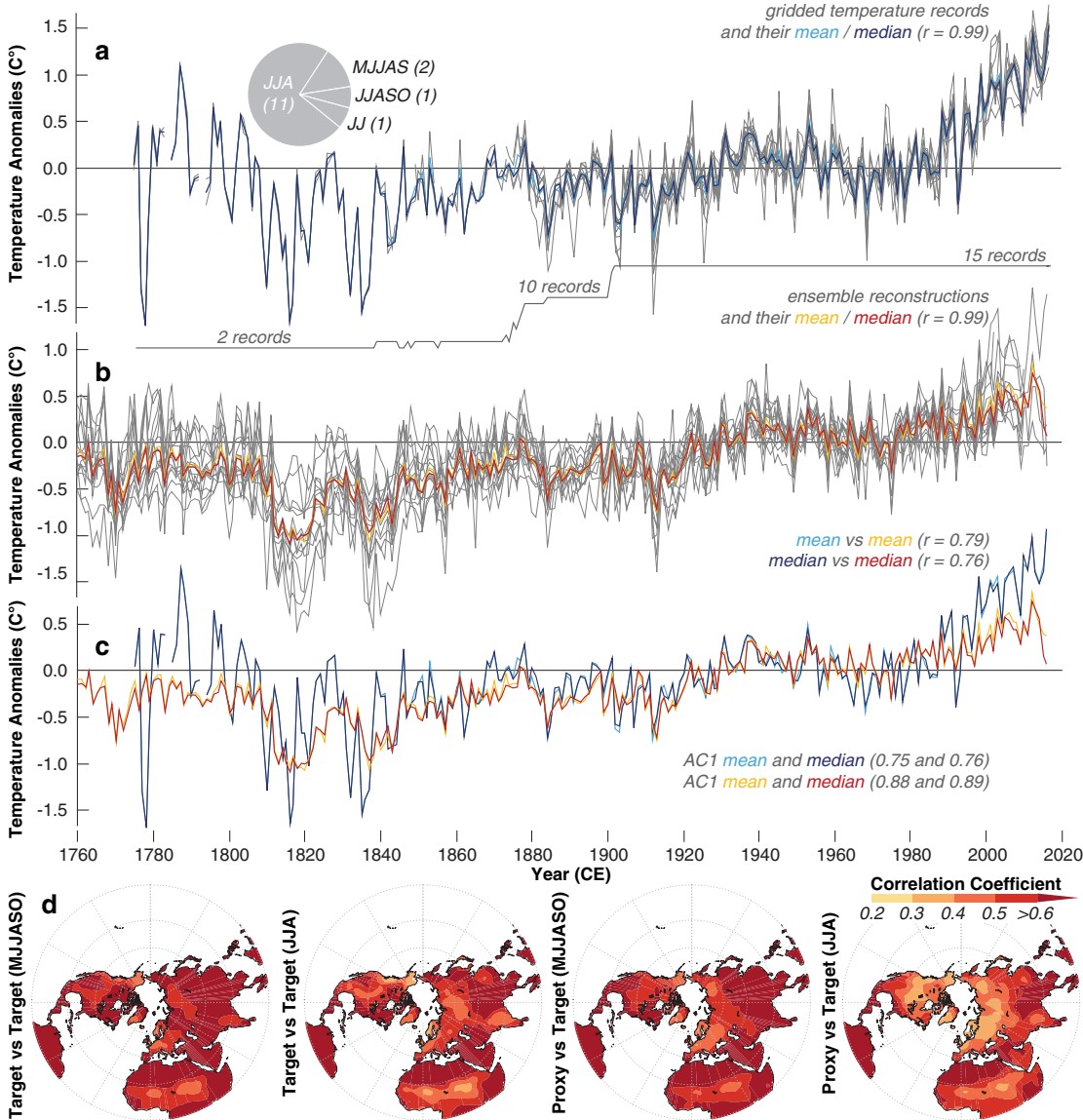

**Fig. 3 Measured and reconstructed temperatures. a** Gridded temperature measurements (grey lines), together with their mean and median time-series (see the "Methods" section for details). The pie chart shows the seasonal averages used, and the lower grey solid line refers to the number of records. **b** Ensemble reconstructions (grey lines), together with their mean and median (orange and red). The mean and median of both, the proxy and target data are statistically similar ($r = 0.99$). **c** Measured and reconstructed temperature means and medians. **d** Spatial field correlations between the mean of all 15 temperature targets and gridded Berkeley data (Target versus Target), and the ensemble reconstruction mean and gridded Berkeley data (Proxy versus Target), calculated over 1794–2016 CE.

0.46–0.54 and RE/CE values drop to 0.20–0.29. In summary, the reconstruction mean reveals significantly positive calibration/verification statistics with a reasonably robust predictive skill for different frequency domains from interannual to centennial.

Relatively warmer measured temperatures prior to circa 1850 CE are possibly biased by the low quality and quantity of early instrumental observations[19], whereas relatively cooler reconstructed temperatures after circa 1990 CE are symptomatic for the 'Divergence Problem' in dendroclimatology[20]: the apparent decoupling between TRW chronologies and rising temperature measurements since around the 1970s[21]. Recent investigations suggest that methodology-induced challenges of proxy-target calibration, proxy network size, end-effects in time-series composition[22], as well as industrial pollution[23], or a combination thereof[20], can explain the 'Divergence Problem'. Three

reconstructions track the measured warming after 1990 very well (R8–R10), and another three ensemble members reveal just slightly lower temperatures compared to those measured between 1990 and 2000 (R7, R11, R14). Evidence for the 'Divergence Problem' is clearest in R1, R5 and R13, which are ~0.4 °C colder than the measured summer temperatures from 1990 to 2000 CE. It should be noted that the end dates of the 15 ensemble reconstructions vary between 2000 and 2016 CE depending on choices made by the 15 groups, which, in addition to the different site compilations, can influence the apparent magnitude of the observed recent proxy-target offset.

Further evidence of the coherence between reconstructed and measured summer temperatures is the degree to which their spatial correlation fields correspond with an equivalent instrumental target-to-target correlation (Fig. 3d). The overall higher

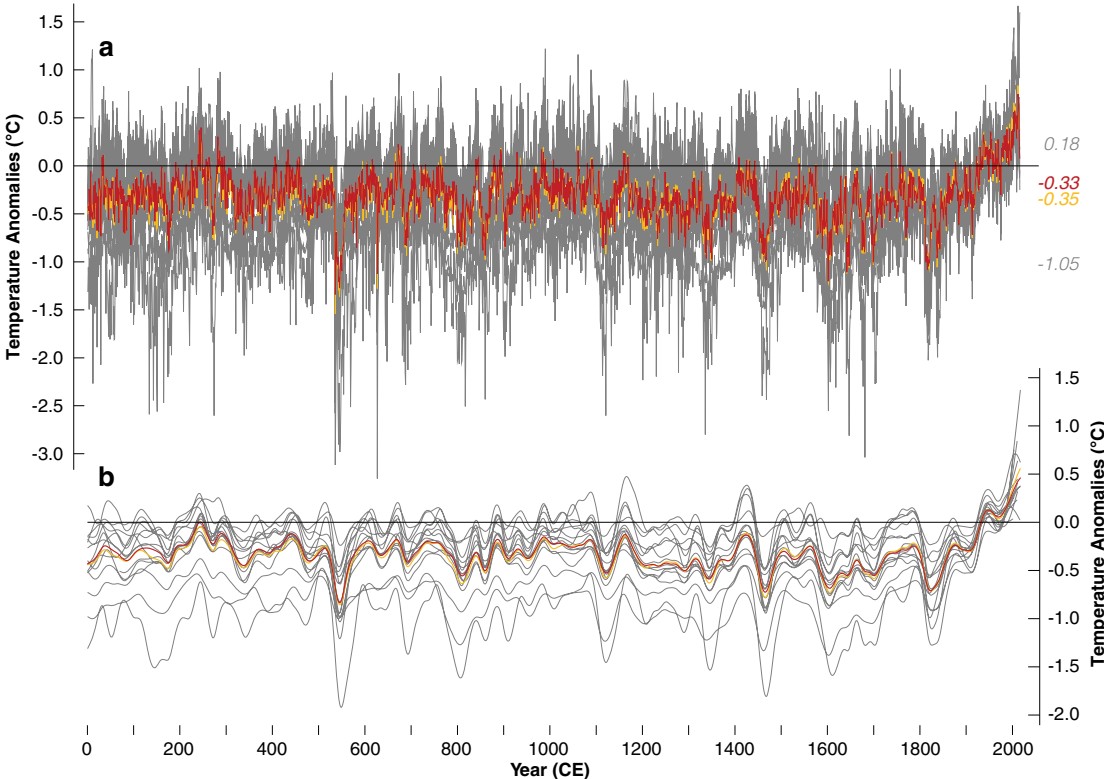

**Fig. 4 Temperature reconstructions. a** Ensemble of 15 reconstructions (grey lines), together with their mean and median (orange and red). Numbers on the right refer to the long-term mean of the maximum, and minimum, as well as mean and median values between 1 and 2016 CE. The mean and median correlate significantly at 0.98 ($p < 0.0001$) and exhibit similar first-order autocorrelation coefficients of 0.71 and 0.70, respectively. **b** The 15 ensemble reconstructions and their mean and median after applying 50-year cubic smoothing spline functions.

spatial field correlations of the May–October (MJJASO) temperature mean compared with the shorter JJA window may result from spatially more heterogeneous summer climate, as well as slightly stronger trends in the longer season. Moreover, the spatial correlation fields are likely dominated by those regions that have similar warming trends and therefore contribute most to the large-scale mean. Since the mid-latitude TRW chronologies are generally located in areas of less intensive recent warming, the use of the entire instrumental network for calibration may be an important factor for the observed recent divergence between warmer measured and colder reconstructed large-scale temperatures (Fig. 3c).

Despite the 'Divergence Problem' in the most recent decades, the current warming trend captured by the reconstruction mean and median since the mid-20th century is unusual in the context of the past two millennia (Fig. 4a). The long-term averages of the ensemble mean and median are −0.33 and −0.34 °C below the 1961–90 CE reference climatology. Three ensemble reconstructions exhibit noticeably lower long-term averages (R4, R7 and R14) (Fig. 4b). The warmest and coldest summer temperature anomalies of the ensemble reconstructions, between 1 and 2016 CE, are in 2012 (mean/median = 0.84/0.75 °C and min/max = 0.22/1.67 °C) and 536 CE (mean/median = −1.54/−1.34 °C and min/max = −3.36/−0.38 °C), respectively (Supplementary Table 3). In nine ensemble reconstructions, six years between 1994 and 2016 CE contain the warmest summers of the Common Era, whereas eight and three reconstructions find 536 and 545 CE as the coldest summers, respectively. Pre-industrial summer warmth is most evident in the late-3rd and early-4th centuries, whereas interannual to decadal cooling mostly follows volcanic eruptions[3]. The overall coldest summer in the R4 reconstruction is 627 CE (−3.61 °C),

which was likely caused by a large volcanic eruption of yet unknown source in 626 CE[24].

In addition to the methodologically induced variance amplification or suppression on different frequency domains, temporal changes in the offset between individual reconstructions are biased by the scaling period used (Supplementary Fig. 1). If scaled over 1961–90 CE, the largest variance offset between the ensemble reconstructions coincides with the onset of the Late Antique Little Ice Age (LALIA) in the mid-6th century[25], whereas the strongest agreement is found during the recent warming from the mid-19th century to 2000 CE. This picture, however, changes when using the coldest 30-year interval from 536–565 CE for scaling (i.e., offset during the LALIA decreases but increases during the 20th century). This scaling experiment not only underscores the influence of different calibration periods but also shows how vulnerable our reconstructions are to the choice and quality of their meteorological target data.

**Discussion**
Despite some indication of late Roman warming (in the latter half of the 3rd century) and Medieval warmth (roughly between the 10th and 12th century)[3], all reconstructions lack obvious signs of long-term orbital forcing[10]. The absence of a pre-industrial cooling trend in our data could be related to a combination of proxy-specific behaviour and methodologically induced constraints[26]. The three coldest periods of the Common Era follow clusters of large volcanic eruptions in the middle of the 6th and 15th centuries, as well as in the early-19th century (Supplementary Fig. 2). Despite considerable mean level offsets, all reconstructions show a clear response to the 24 strongest volcanic eruptions of the past two millennia[27] (Supplementary Fig. 3a).

Post-volcanic cooling relative to the previous 10 years is most pronounced in the first two years (Supplementary Fig. 3b).

Comparison of our ensemble reconstructions against the only reconstruction of extra-tropical NH summer temperature variability over the past 1400 years that exclusively uses MXD measurements[28], reveals a high level of coherency (Supplementary Fig. 4a). The independent reconstructions correlate at 0.44 over their common period (600–2002 CE), and high- to low-frequency variability is most similar during the recent five centuries ($r = 0.63$, 0.60 and 0.55 when calculated back to 1600, 1500 and 1400, respectively). Furthermore, the TRW-based and MXD-based reconstructions show similar mid-frequency variance changes over time (Supplementary Fig. 4b). Steadily increasing differences between the reconstructions back in time are likely caused by a substantial variance reduction in the MXD record (Supplementary Fig. 4c, d), which almost certainly results from the decrease in the number of MXD site chronologies from 15 to only three prior to 1363 CE. This issue has been investigated at different spatiotemporal scales[29]. Moreover, AC1 of the MXD data is 0.42, compared with 0.73 in the TRW data (600–2002 CE), underlining the extent that biological memory influences our ensemble reconstructions. If not adequately removed, the TRW-specific mid-frequency amplification due to biological memory[7–9], can affect the interannual behaviour of climate reconstructions. One demonstrable example of this possible bias is the faster recovery of the MXD-based summer temperatures following the negative forcing of volcanic eruptions (Supplementary Fig. 5). In line with the TRW data, the MXD data show their strongest cooling response to volcanic forcing in the first year after eruptions, but reveal substantially higher temperatures in the second year.

Our double-blind experiment created 15 different ensemble reconstructions (R1–R15) with different spectral and statistical properties (Figs. 2 and 5). R4 exhibits the lowest mean, the highest standard deviation, the largest industrial warming, and the strongest post-volcanic cooling. R8 contains by far the lowest AC1, and the smallest difference between temperature means before and after 1850 CE. By including instrumental temperature measurements in the reconstruction, R10 obviously reveals the strongest agreement with the mean of the 15 slightly different gridded instrumental target time-series ($r = 0.81$ from 1794 to 2000 CE). This approach also produces the lowest 5-year post-volcanic cooling ($-0.48°$ with respect to the 10-year pre-eruption period). R14 displays the highest degree of long-term persistence, which was calculated in addition to the reconstructions' simple autocorrelation structures and power spectra (Supplementary Figs. 6–9). Local monthly temperature measurements typically exhibit Hurst exponents ($H$) of 0.55–0.75 during the 20th century[30] ($H > 0.5$ shows the presence of long-term memory in a system), whereas the NH summer mean contains more persistence ($H = 0.76$). With the exception of R5 and R8, most ensemble reconstructions exhibit more long-term persistence than the observational data (see Supplementary Fig. 9 for details of the reconstructions' individual power laws).

Our community-driven ensemble experiment demonstrates how an investigator's decision-making process influences the behaviour and characteristics of climate reconstructions. Although an objective ranking of reconstructions is not possible since the true temperature history of the Common Era is unknown, we can make the following observations: Those reconstructions that correlate strongly with an independent instrumental target and contain relatively low AC1, such as the reconstruction mean and median, likely reflect better methodological choices in terms of signal preservation (Supplementary Fig. 10). Since R10 integrates instrumental temperature measurements during the calibration period, this reconstruction is not entirely independent of the

target data. Note, too, that R9 and R10 likely both underestimate low-frequency variability due to the individual series detrending applied[16], emphasising that the selection of detrending methods should not be based on calibration period statistics alone. In using five sites only, R7 evidently captures temperature variability across the smallest area of the NH extra-tropics. Those reconstructions that selected 2-month or 5-month seasonal windows are possibly under-representing or over-representing the JJA period that seems optimal for TRW formation at most of the nine sites. Last but not least, we believe that AC1 values well above 0.7 are indicative of too much retained biological memory, e.g., R2, R6 and R7, R11 and R12, and R14 and R15 (Supplementary Fig. 10).

Our ensemble approach not only confirms existing evidence of summer temperature variability during the past millennium[28,31,32] but also provides insights into the range of reconstruction variability before medieval times. Comparison with the PAGES19 multi-proxy product[15], which reflects global annual mean temperatures between 1 and 2000 CE (Supplementary Fig. 11), reveals reduced large-scale cooling between the mid-13th century and around 1900 CE in the TRW-only reconstructions. Although this finding is consistent with previous observations of differing pre-industrial cooling trends between tree rings and lower-resolution temperature proxies[26], it is not clear whether this offset arises from methodological choices, the composition of the proxy network, or both. Despite a different temperature amplitude on decadal to centennial time scales in the TRW reconstructions compared with the re-scaled PAGES19 product (Supplementary Fig. 11c), the strong positive correlation coefficients indicate agreement in the trajectory of Common Era temperature changes. PAGES19 correlates with our reconstruction mean and median at 0.60 and 0.62 from 1 to 2000 CE, respectively. The relationship remains stable when using the first half of the Common Era ($r = 0.62$ and 0.58), but increases to 0.73 and 0.74 from 1001–2000 CE. Higher correlation coefficients are obtained after 50-year low-pass filtering the time-series (Supplementary Fig. 11c). The PAGES19 reconstruction is characterised by an overall higher AC1 of 0.88 compared to 0.70 in the TRW-based reconstructions (1–2000 CE), which is likely due to the predominance of lower-resolution proxies in the first millennium of the Common Era and the global coverage of the PAGES19 reconstruction.

In conclusion, we advocate for the routine use of ensemble techniques to develop more consensual climate reconstructions and better quantify their uncertainties. Though it likely underestimates variance, we consider the reconstruction mean the most robust NH temperature estimate and suggest using the upper and lower ranges of the 15 individual reconstructions as uncertainty ranges caused by the methodological choices of the investigators. Whenever possible, collaborative endeavours should focus on the development of multi-millennia-long tree-ring chronologies (especially from MXD) in those regions of both hemispheres that are presently under-sampled. Last but not least, we call for the improvement of wood anatomical and isotopic records, and the combined assessment of different tree-ring parameters in advanced multivariate fusion approaches.

## Methods

**Tree-ring proxy and instrumental target data.** We compiled and analysed updated versions of all the existing summer temperature-sensitive TRW chronologies from the NH that span the entire Common Era[3]. These include regional site chronologies of living trees and relict materials from the Great Basin (GTB) in the western United States[33], the southern Colorado Plateau (SCO) in the western United States[34], Quebec and Labrador (QUL) in Canada[35], Northern Scandinavia (NSC) in Sweden and Finland[36,37], the Austrian Alps (ALP) in Austria[25,38], the Yamal Peninsula (YAM) in northern Russia[39], the Taimyr Peninsula (TAI) in northern Russia[40], the Altai Mountains (ALT) in southern Russia[25], and Northern Yakutia (NYA) in northeastern Russia[41]. The mid-latitude TRW collections from the western United States (GTB and SCO), Austrian Alps (ALP), and Russian Altai (ALT) are from upper treeline pine and larch ecosystems. The collections from

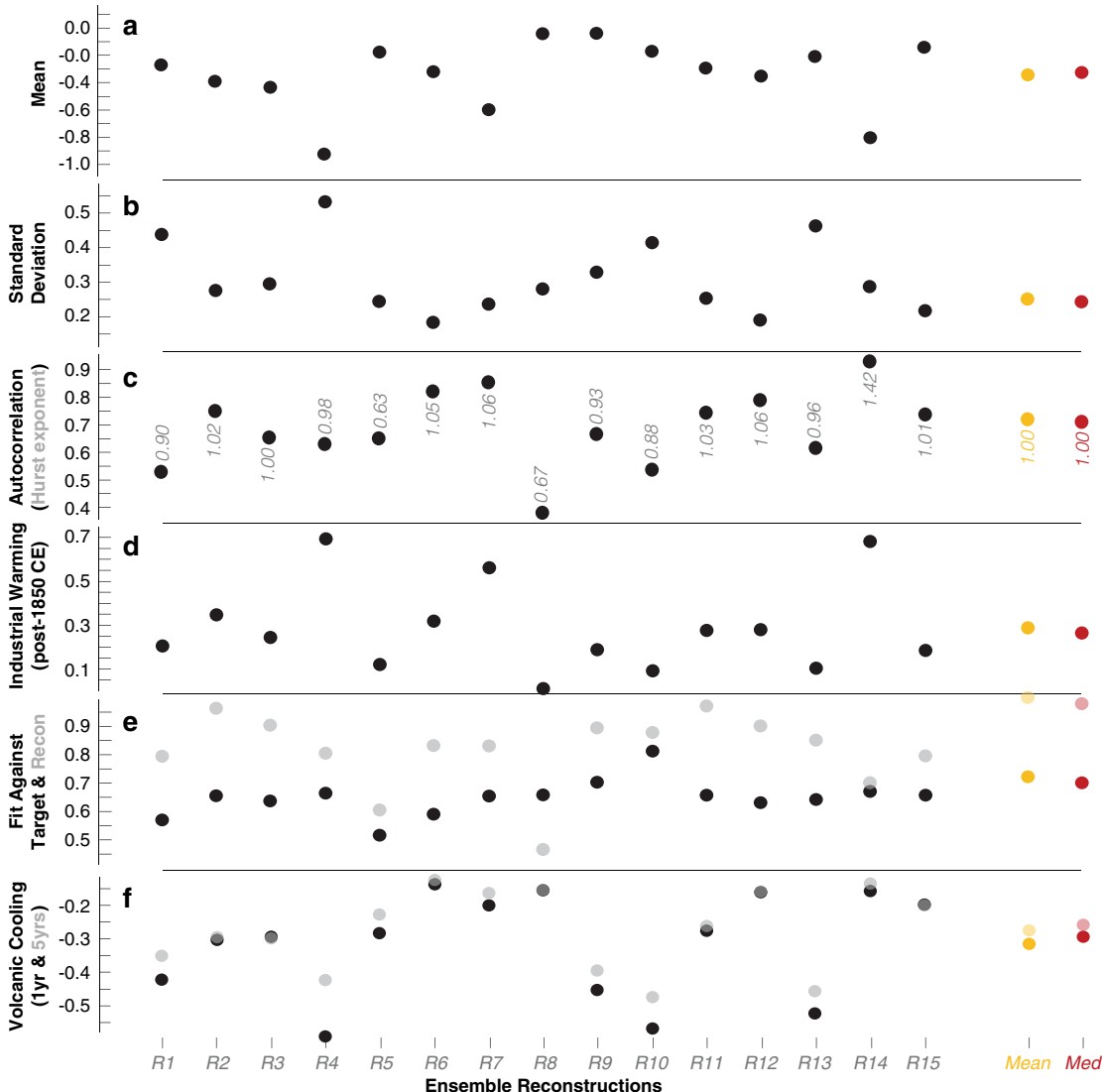

**Fig. 5 Reconstruction characteristics. a–d** Mean, standard deviation, first-order autocorrelation (and highest Hurst exponent $H$), and the difference between industrial and pre-industrial temperature means (after and before 1850 CE) of the 15 ensemble reconstructions (R1–R15), as well as their mean and median (orange and red). **e** Correlation coefficients of the 15 ensemble reconstructions against the gridded instrumental target mean (Target Fit; black) and against the reconstruction mean (Recon Fit; opaque). **f** One (black) and 5-year (opaque) post-volcanic cooling relative to the 10 pre-eruption years (see Supplementary Fig. 3 for the 24 eruption years used). All statistics refer to 72–2000 (or 1794–2000 CE when considering the mean of the instrumental temperature records).

Quebec and Labrador (QUL; spruce), northern Scandinavia (NSC; pine), and northern Siberia (YAM, TAI and NYA; larch) are from the northern boreal forest. The number of TRW samples from each collection, either in the form of increment cores or stem discs, ranges from 224 (SCO) to 2725 (QUL). Each regional TRW dataset has been updated into the 21st century (Fig. 1). The majority of TRW measurements was derived from remnant snags and subfossil wood remains (i.e., there is a much larger proportion of samples from relict materials versus living trees). The annual cross-dating precision of all TRW measurements within each collection has been independently confirmed by the detection of cosmogenic radiocarbon events in 774 and 993 CE, i.e., abrupt changes in the Earth's atmospheric radiocarbon ([14]C) abundance[6]. For each of the nine regional TRW datasets, neither the number of TRW series nor the mean segment length and the mean series age is stable over time[3]. The 15 independent participating groups all based their reconstruction on a combination of these nine regional TRW datasets, but each used a slightly different gridded instrumental temperature dataset as a reconstruction target. These targets include four seasonal windows of 2, 3 and 5 months between May and October (Fig. 2). The start and end dates of the 15 target datasets vary between 1750 and 1901 CE and 2000 and 2016 CE, respectively. Restricted to the NH, nine latitudinal bands between the equator and the pole, as well as four regional means were selected as spatial target domains. The 15 groups extracted data from various versions of the gridded Berkeley[42], CRU[43–47] and HadCRU[48] products (see Fig. 2 and Supplementary Table 1 for an overview).

**Ensemble reconstructions**. In the first step of data screening, nine groups did not remove any of the raw TRW measurement series (R1–R3, R6 and R7, R9 and R12–R14). R4 removed all TRW series that correlate at $r < 0.2$ with the site master chronology, R5 removed all TRW series <40 years, R8 removed all TRW series <100 years, R10 used only 200 TRW series per site based on an optimal number of data points and evenness of coverage, R11 removed <10 duplicate TRW series amongst all datasets, and R15 removed all TRW series <50 years. Secondly, seven groups used recent versions of the Signal-Free Regional Curve Standardisation (SF RCS)[49] for TRW detrending (R2–R6, R11 and R14). R7 and R13 used traditional RCS detrending[18], R1 developed an ensemble of 16 different RCS chronologies per site[3], R8 applied the adaptive regional growth curve method for detrending (ARGC)[50], R9 used negative exponential curves for individual TRW series detrending[16], R10 used cubic smoothing splines with 50% frequency cut-off at 2/3 of the individual TRW series length, R12 used age band decomposition for detrending (ABD)[17], and R15 detrended all TRW series with non-significant negative slopes by subtracting their mean and applied SF RCS on the remaining data. Based on site-specific temperature signals, nine groups included all nine TRW chronologies in their final large-scale reconstruction (R2 and R3, R5 and R8–R13). R1 and R6 excluded data from the western United States and used seven chronologies (QUL, NSC, ALP, YAM, TAI, ALT, NAY), R4 and R15 excluded two sites from central and eastern Siberia used another combination of seven chronologies (GTB, SCO, QUL, NSC, ALP, YAM, ALT), and R7 only considered information from five TRW chronologies in their final reconstruction (GTB, NSC, ALP, YAM,

ALT). Eleven groups selected June–August mean temperatures as the optimal season for reconstruction (R1 and R2, R4, R6, R8–R14), whereas two groups selected the window from May to September (R3, R7), and R5 and R15 used the mean June–July and June–October temperature as a target, respectively. None of the groups used the same large-scale temperature target data, which varied been gridded products, spatial domains and time periods. For calibration, three groups used simple scaling (R1 and R2, R11) or regression (R4, R6, R12) technique[51], another three groups used a nested principal component regressions (PCR)[52,53] approach (R3, R7, R9), R13 and R15 used nested composite plus scaling (CPS)[32,54], R5 used PCR, R8 used analogue frequency regression (AFR)[55], R10 used scaling and splicing with the target, and R14 used a gaussian process regression (GPR)[56] for reconstruction (see the online Supplementary Information for more details on the network compilation, the reconstruction procedure, the calibration/verification trials, and the associated error bars). Please note that an archive of underlying code is not available for this ensemble experiment, because the 15 different groups used a wide range of techniques and applied different software programmes for the various methodological steps involved.

**Persistence measure.** To avoid the effects of linear deterministic trends in the time-series when estimating their degree of long-term persistence[57], we applied both detrended fluctuation analysis (DFA2)[58] and wavelet analysis (WT2)[30,59,60]. For long-term persistent records the DFA2 fluctuation function $F(s)$ and the WT2 fluctuation function $G(s)$ show the same power-law behaviour, but on different time scales: WT2 on short time scales and DFA2 on longer time scales. The deviation of the Hurst exponent ($H$) from 0.5 (white noise) quantifies the strength of the long-term persistence of the record. In both DFA2 and WT2 one measures the variability of a record by studying the fluctuations in segments of the record as a function of the segment length $s$. In the first step, the record $\{y_i\}$, $i = 1, 2,…, N$, is divided into $v$ nonoverlapping windows of length $s$. In DFA2 in each segment, one considers the cumulated sum $Y_i$ of the data and calculates its variance $F_v^2(s)$ around the best polynomial fit of order 2. Then one averages the variances $F_v^2(s)$ of all segments and takes the square root to arrive at the desired fluctuation function $F(s)$. For long-term persistent records, it can be shown, that $F(s) \sim s^h$ for about $8 < s < N/4$, with the Hurst exponent $H = (2-\gamma)/2$ (gamma is the power-law exponent that describes the decay of the autocorrelation function for long-term persistent data). At very short timescales the power-law scaling behaviour of $F(s)$ does not hold, because of the crossover inherent to the method. WT2 offers itself as a complement to DFA2 on short time scales. In WT2 one calculates, in each segment $v$ the mean value $\bar{y}_v$ of the data and considers the linear combination $\Delta_v^{(2)} = \bar{y}_v(s) - 2\bar{y}_{v+1}(s) + \bar{y}_{v+2}$ . Then one average $(\Delta_v^{(2)})^2$ over all segments $v$, takes the square root and multiplies by s to arrive at the desired fluctuation function G(s). For long-term persistent records $G(s) \sim s^h$ from $s = 1$ to about $s = N/40$. Accordingly, for long-term persistent records $F(s)$ and $G(s)$ show the same power-law behaviour, although on complementing time scales: WT2 on short time scales and DFA2 on longer time scales. Hurst exponents well above 1/2 signify a strong long-term persistence in the record. For behaviour that is in reality a simple autoregressive process of first order (AR1), lag-1 autocorrelation (AC1) can describe the persistence properties of a record if no external trends are present. However, the analyses of instrumental data show, as mentioned above, that temperatures are long-term persistent. This means that a temperature data point $y_i$ depends directly not only on its predecessor $y_{i-1}$, as it would be the case for AR1, but on all preceding data points $y_{i-1}…y_1$. The further back in time a preceding data point is, the lower its influence on $y_i$, but in the mathematical model it never vanishes. If a record is long-term persistent, it often implies, depending on the specific parameters, that long-lasting deviations from the long-term mean are more probable than they would be if one only assumes an AR1 persistence in the record. Furthermore, an AC1 or full autocorrelation analysis can also be incorrect if external trends are present in the data. In contrast, DFA2 and WT2 are not influenced by external linear trends, since linear trends are eliminated by construction.

## Data availability
All tree-ring width data used in this study are freely available at the NOAA National Centers for Environmental Information (NCEI): https://www.ncdc.noaa.gov/paleo-search/study/33215

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

## Acknowledgements

R. Neukom kindly provided the re-scaled PAGES 2k data. U.B. and J.E. received funding from SustES: Adaptation strategies for sustainable ecosystem services and food security under adverse environmental conditions (CZ.02.1.01/0.0/0.0/16_019/0000797), and the ERC project MONOSTAR (AdG 882727). C.C., S.G. and M.S. received funding from the SNF Sinergia project CALDERA (project no. 183571). S.C. acknowledges support from US National Science Foundation grants 1737918, 1939916 and 1939956.

## Author contributions

U.B. designed the study with input from P.J.K. and J.E. The regional tree-ring width datasets were developed by D.A., É.B., O.V.C.(S.), F.G., M.K.H., A.V.K., V.S.M., K.N., F.R. and M.W.S., whereas U.B., K.Al., K.An., A.B., S.C., C.C., J.G., S.G., J.G., B.G., S.H., P.Ho., P.Hu., P.J.K., J.L., W.J.-H.M., C.O., K.S., A.R.S., M.S., S.S.G., E.T., A.T., V.T., J.W., R.W., B.Y., G.X. and J.E. contributed to the ensemble reconstructions and their interpretation. U.B. wrote the paper together with C.O., P.J.K., P.C. and J.E.

## Competing interests

The authors declare no competing interests.

## Additional information

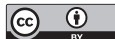

[1]Department of Geography, University of Cambridge, Cambridge, UK. [2]Swiss Federal Research Institute (WSL), Birmensdorf, Switzerland. [3]Global Change Research Centre (CzechGlobe), Brno, Czech Republic. [4]Department of Geography, Faculty of Science, Masaryk University, Brno, Czech Republic. [5]School of Ecosystem and Forest Sciences, University of Melbourne, Richmond, Australia. [6]ARC Centre of Excellence for Australian Biodiversity and Heritage, University of NSW, Sydney, Australia. [7]School of Geography, Development, and Environment and Laboratory of Tree-Ring Research, University of Arizona, Tucson, AZ, USA. [8]Department of Biology, Chemistry and Geography, University of Quebec in Rimouski, Rimouski, QC, Canada. [9]Department of Geography, Université du Québec à Montréal, Montréal, QC, Canada. [10]GEOTOP, Université du Québec à Montréal, Montréal, QC, Canada. [11]Centre d'Études Nordiques, Université Laval, Québec, QC, Canada. [12]Institute of Geography, Friedrich-

Alexander-University of Erlangen-Nürnberg, Erlangen, Germany. [13]School of Statistics, University of Minnesota, Minneapolis, MN, USA. [14]Institute of Ecology and Geography, Siberian Federal University, Krasnoyarsk, Russia. [15]Université Clermont-Auvergne, Geolab UMR 6042 CNRS, Clermont-Ferrand, France. [16]Institute for Environmental Sciences, University of Geneva, Geneva, Switzerland. [17]GREMA and Forest Research Institute, Université du Québec en Abitibi-Témiscamingue, Amos, Canada. [18]Aix Marseille University, CNRS, IRD, INRA, Coll France, CEREGE, Aix-en-Provence, France. [19]Department of Physical Geography, Bolin Centre for Climate Research, Stockholm University, Stockholm, Sweden. [20]Natural Resources Institute Finland, Rovaniemi, Finland. [21]Laboratory of Tree-Ring Research, University of Arizona, Tucson, AZ, USA. [22]Department of Earth and Planetary Sciences, Harvard University, Cambridge, MA, USA. [23]Sukachev Institute of Forest SB RAS, Krasnoyarsk, Russia. [24]Potsdam Institute for Climate Impact Research (PIK), Potsdam, Germany. [25]Institute of Humanities, Siberian Federal University, Krasnoyarsk, Russia. [26]Department of Geography, University of Innsbruck, Innsbruck, Austria. [27]McDonald Institute for Archaeological Research, Cambridge, UK. [28]Department of Geography, Johannes Gutenberg University, Mainz, Germany. [29]Department of Earth Sciences, Goteborg University, Goteborg, Sweden. [30]Department of Earth & Climate Sciences, San Francisco State University, San Francisco, CA, USA. [31]Department of Earth Sciences, University of Geneva, Geneva, Switzerland. [32]Department F.-A. Forel for Environmental and Aquatic Sciences, University of Geneva, Geneva, Switzerland. [33]Department of Geography, Environment and Society, University of Minnesota, Minneapolis, MN, USA. [34]Department of Atmospheric and Environmental Sciences, University at Albany (SUNY), Albany, NY, USA. [35]Key Laboratory of Desert and Desertification, Northwest Institute of Eco-Environment and Resources, Chinese Academy of Sciences, Lanzhou, China. [36]CAS Centre for Excellence in Tibetan Plateau Earth Sciences, Chinese Academy of Sciences, Beijing, China. [37]Qinghai Research Centre of Qilian Mountain National Park, Academy of Plateau Science and Sustainability and Qinghai Normal University, Xining, China. [38]School of Earth and Environmental Sciences, University of St Andrews, Scotland, UK. [39]Lamont-Doherty Earth Observatory of Columbia University, Palisades, NY, USA. [40]State Key Laboratory of Cryospheric Sciences, Northwest Institute of Eco-Environment and Resources, Chinese Academy of Sciences, Lanzhou, China. ✉email: ulf.buentgen@geog.cam.ac.uk

