## [Peer Review File · Nature Communications]

REVIEWER COMMENTS, first round -

Reviewer #1 (Remarks to the Author):

Dear Authors, Dear Editor,

The study of NCOMMS-20-46730-T presents a community-driven ensemble approach employing independently developed Northern Hemisphere, summer temperature reconstructions from a network of regional tree-ring width datasets. Fifteen independent groups with great expertise in dendroclimatological methods developed reconstructions using the same set of tree-ring width database but applying different methodologies. Based on the detailed descriptions of the Supplementary information it is clear that each group applied a sound and well-established methodology. The 15 independent reconstructions were evaluated. I think this small-ensemble is really appropriate to illustrate the "researcher bias" (such as different selection criteria, detrending methods) in the resulting reconstructions at multi-millennial and large spatial scales.

Although it is a well-known fact that the different detrending methods will produce different index series (for instance the degree of the retained low-freq variance), however I've found it extremely exciting to see the differences among the ensemble members which, without exception, have produced excellent verification statistics.

The presented results clearly support the conclusions and claims. The applied methodology is sound and the results are interesting. The illustration material is of high-quality. I have not found any flaws in the data analysis which could bias the conclusions.

Authors close the paper with a methodological recommendation proposing the routine use of non-hierarchical ensemble techniques, rather than following the traditional single-solution pathways. It may have be of interest to other groups in the dendro-community and even in the wider field of proxy-paleoclimatology.

I have only two minor comments both are related to the Supplementary.

-line 973 The year should be corrected in the reference at the end of the line.

-line1062: Please, explain what gamma means in the formula of the Hurst exponent.

Reviewer 2:

[see next page]

Review of
**An ensemble approach to Common Era temperature
reconstructions from tree rings**
by Büntgen et al.

Recommendation: *Major revisions*

Summary The paper investigates the impact of methodological choices on global temperature reconstructions based on tree-ring width proxies. It does so by enlisting a large consortium of authors implementing various approaches, and comparing the effect of these approaches on various metrics of global temperature change (extremes, amplitude, spectral properties). The topic is important, the study design is sound and the figures of high quality; yet, the manuscript itself falls flat in several respects. I recommend revisions to make the paper more broadly useful to the climate community at large, and these suggestions are numerous enough to qualify as major revisions.

1 General Comments

After a careful read, I see three main issues that limit the current usefulness of the paper.

1.1 Dendrocentrism

Given the centrality of dendrochronology in high-resolution paleoclimatology, and given the limited traceability of many methodological decisions (e.g. detrending) in the published products, the approach taken here is very well conceived. However, I think the paper would be of broader interest if it were written for a broader audience. As it stands, too many assumptions are made about the reader's familiarity with the ins and outs of dendrochronology, particularly its jargon. To 99.9% of scientists, detrending means one thing; to dendrochronologists, it means removing the non-climatic component of growth – yet the reader would not know that from the current exposition. The same applies to a number of other jargony terms (e.g. “disc sample”). I recommend aiming the paper at a wider audience, not just the tiny fraction of the human population that speaks that lingo. I believe that is congruent with the goals of a non-disciplinary journal like Nature Communications. Surely there is a way to explain jargon in the methods section.

1.2 Weak conclusions

The paper's conclusions are rather weak, given the involvement of such a distinguished group of experts. As far as I can tell, the message is: methodological choices matter; one should use a similar approach (somewhat pompously called "non-hierarchical ensemble techniques") to quantify their impact. It's hard to disagree with that. In fact, the real question is why it has taken the community so long to get to the point of truly collaborating in this manner.

Yet it is excessive to claim that the paper “improve(s) our understanding of NH summer temperature variability before medieval times” (L229). What it does is shed light on the impact of the various choices, but beyond a dry description of which year is warmest/coldest in each reconstruction, there is (1) very little analysis of why the results end up being what they are and (2) what to do about it. That is, if a scientist outside the author group decided to re-use any of the reconstructions for a particular application, which should they use? If (as seems likely), there is no reason to prefer one reconstruction over another, how should they be optimally combined? One could use the whole plume, of course, but since the authors put forth the mean and median, presumably those are their favored options? If so, why?

The arithmetic mean is optimal in the context of the central limit theorem, which assumes independent variations. Is that the case here? If not, what is the rationale for using the mean? Is there any sense in a weighted mean, as in Bayesian model averaging? It would behoove the paper to provide a rationale/interpretation for a reduced timeseries like the mean or median. In such a small ensemble, I worry that those reduced timeseries are effectively a referendum on which choice was made most often.

The paper also makes a lightning-fast comparison to the PAGES 2k reconstruction ensemble of *Neukom et al.* [2019]. I think many readers would be interested in a more in-depth comparison.

Finally, there is throughout the paper an emphasis on the persistence properties of the reconstructions, whether measured by the lag-1 autocorrelation (called ‘AC1’ in the manuscript, though I don’t believe this notation is defined anywhere) or the Hurst exponent. What to make of these metrics? It is good to document them, but I did not see them used in a meaningful way. On L1324, it is said "Interestingly, on medium time scales between six and about 40 years the Hurst exponents are 1.325 1.42 (DFA2) and 1.44 (WT2)." Why is that interesting?

In summary, the paper goes to great lengths to document the methods and their impacts on various metrics, but falls short of making any meaningful scientific conclusions out of it. For instance, the response to volcanic cooling has been suggested as a way to probe transient climate sensitivity [*Merlis et al.*, 2014]. Maybe this ensemble could be used for that purpose, quantifying the distribution of TCRs that results from the methodological uncertainty? That would be a tad more interesting than the current, dry list of statistics.

1.3 Reproducibility

The methods section is rather lengthy and repetitive. It is detailed enough to get a general idea of what was done, too detailed to be properly digested by a sane reader, and not detailed enough to reproduce the results. To do that, in 2020 (soon to be 2021), it is absolutely essential that the authors share their code, preferably on a modern platform like GitHub.

The description also makes it challenging to understand the causes of the differences between the various reconstructions, since several chunks of texts are common. Table S1 is a lot easier to parse, and maybe the methods section could be organized as a narrative explaining the various abbreviations in the table, and a brief rationale for each choice. That is, instead of focusing on what each reconstruction did, it would focus the choices available in each "column", and explain their relative merits/rationales.

2 Editorial Comments

- L93** “this technique has only rarely been considered for multi-proxy temperature reconstructions”. I fail to see how *Mann et al.* [2008] (reference 5) exemplifies that ; there are no ensembles in there.
- L126** “A wide range of scaling and regression techniques was applied”. Are they really that different in the grand scheme of things? Again, referring to Table S1, full of jargony abbreviations like “nested PCR”, is assuming a lot of the audience. Having a method-centric methods section, as opposed to a group-centric one, would solve that.
- L134-135** “their mean and median track the instrumental measurements remarkably well between the end of the 19th century and around 1990 CE”: is that really a surprise, given that this is the calibration period?
- L135-136** “Proxy-target correlations are 0.76 and 0.79 for the reconstruction mean and median, respectively”. For series with this much autocorrelation, it is not out of this world. Please compare those numbers with *Neukom et al.* [2019], Table 1.
- L150** “the ‘Divergence Problem’ did not exist prior to the Anthropocene”. Which Anthropocene are we talking about? The one that began in 1784 with James Watt’s invention of the steam engine, as proposed by Crutzen? Or the one that began in 1945 with the detonation of the Trinity bomb, which ushered the start of the Atomic Age? Or the Anthropocene that started thousands of years ago with the start of agriculture? You get the point: the start of the Anthropocene is a disputed notion. Why not use a calendar date, to lift ambiguity?
- L157** “spatial correlation fields of the ensemble mean spatial correlations with observed temperatures resemble the geographical extent obtained from a simple instrumental target-to-target comparison”. A tad abstruse. Suggest rewriting for clarity.
- L174-176** “Interestingly, six different years between 1994 and 2016 CE were identified by nine individual reconstructions as the warmest summers, whereas eight and three reconstructions define 536 and 545 CE as the coldest summers, respectively.”. I do not find this trivia particularly interesting. What is the dynamical significance?
- L186** “This side experiment underlines the impact of different scaling periods, and provides further evidence of research bias that can override proxy information.” This is interesting. Do the authors have any explanation for this sensitivity? Any recommendations for the most appropriate scaling choice?
- L196** “The TRW-based cooling signal, however, lasts least one decade, and appears robust”. Robust but demonstrably excessive, as the comparison to MXD-based reconstructions shows, only a few lines below. I fail to see the point of emphasizing this long-term signal, given what we know of the biological memory embedded in TRW (cf L209).
- L209** this memory has major implications for comparison with simulated volcanic cooling, as shown by *Stoffel et al.* [2015] and more recently by *Zhu et al.* [2020]. Given the prominence given to this sort of SEA exercise in AR5, I think it’s worth reminding the reader (again, most likely not a dendroclimatologist) that ring width is misleading at those scales, and that this is a known issue.

- L215** “of diverse statistical properties” → “with diverse statistical properties”, perhaps?
- L226** “all ensemble reconstructions contain more long-term persistence than the observational evidence”. Again, given the biological memory of TRW, is this a noteworthy result?
- L229** “Our findings [...] improve our understanding of NH summer temperature variability before medieval times”. They really only improve our understanding of this particular set of reconstructions. More work needs to be done to explain NH summer temperature variability.
- L241-243** I wonder if the authors will venture to say something about the relative merits of TRW vs MXD reconstructions, and if a similar approach as used here is necessary for MXD-based reconstructions as well.
- Fig 1** the map requires an electronic microscope to consult. Suggest moving the map to an upper panel and squeezing the rest at the bottom. The rightmost panel makes it hard to judge how high the bars are by want of gridlines. A simple y grid aligned with major tick marks would solve that.
- Fig 4F** what is the list of volcanic events used here? It does appear rather consequential in such comparisons.
- L414 and others** “with the later normally containing” → with the latter normally ...
- L1054** “The deviation of the Hurst exponent h from 0.5 (white noise) quantifies the strength of the long-term persistence of the record.” this sentence is repeated verbatim in L1069. One will suffice.
- Table S3** this is not easy to parse. Suggest using shades of blue/red to denote cold and warm extremes, so the eye can grasp them at once.
- Fig S2** (right) the 1809 eruption is not highlighted, yet it clearly seems to be kickstarting the cooling.
- Fig S3** seems rather useless in view of the memory artifact illustrated in Fig S5.
- Fig S6** the top number on the y axis should be 1.0, not 0.1.
- Fig S7** The periodogram is a dreadful spectral estimator, being inconsistent. I wonder why the authors did not use a more reasonable one, like the Multi-Taper Method [*Thomson*, 1982], which might lump together some of these peaks. Also, the significance with respect to an appropriate null (AR(1), for instance) is not given.

JULIEN EMILE-GEAY

References

- Mann, M. E., Z. Zhang, M. K. Hughes, R. S. Bradley, S. K. Miller, S. Rutherford, and F. Ni (2008), Proxy-based reconstructions of hemispheric and global surface temperature variations over the past two millennia, *Proceedings of the National Academy of Sciences*, *105*(36), 13,252–13,257, doi:10.1073/pnas.0805721105.
- Merlis, T. M., I. M. Held, G. L. Stenchikov, F. Zeng, and L. W. Horowitz (2014), Constraining transient climate sensitivity using coupled climate model simulations of volcanic eruptions, *Journal of Climate*, *27*(20), 7781–7795, doi:10.1175/JCLI-D-14-00214.1.
- Neukom, R., et al. (2019), Consistent multidecadal variability in global temperature reconstructions and simulations over the common era, *Nature Geoscience*, *12*(8), 643–649, doi:10.1038/s41561-019-0400-0.
- Stoffel, M., et al. (2015), Estimates of volcanic-induced cooling in the northern hemisphere over the past 1,500 years, *Nature Geoscience*, *8*, 784 EP, doi:10.1038/ngeo2526.
- Thomson, D. J. (1982), Spectrum estimation and harmonic analysis, *Proc. IEEE*, *70*(9), 1055–1096.
- Zhu, F., J. Emile-Geay, G. J. Hakim, J. King, and K. J. Anchukaitis (2020), Resolving the differences in the simulated and reconstructed temperature response to volcanism, *Geophysical Research Letters*, *47*(8), e2019GL086,908, doi:10.1029/2019GL086908.

Reviewer #1

The study of NCOMMS-20-46730-T presents a community-driven ensemble approach employing independently developed Northern Hemisphere, summer temperature reconstructions from a network of regional tree-ring width datasets. Fifteen independent groups with great expertise in dendroclimatological methods developed reconstructions using the same set of tree-ring width database but applying different methodologies. Based on the detailed descriptions of the Supplementary information it is clear that each group applied a sound and well-established methodology. The 15 independent reconstructions were evaluated. I think this small-ensemble is really appropriate to illustrate the “researcher bias” (such as different selection criteria, detrending methods) in the resulting reconstructions at multi-millennial and large spatial scales. Although it is a well-known fact that the different detrending methods will produce different index series (for instance the degree of the retained low-freq variance), however I’ve found it extremely exciting to see the differences among the ensemble members which, without exception, have produced excellent verification statistics.

We are thankful for this positive synthesis of our work.

The presented results clearly support the conclusions and claims. The applied methodology is sound and the results are interesting. The illustration material is of high-quality. I have not found any flaws in the data analysis which could bias the conclusions.

Thank you.

Authors close the paper with a methodological recommendation proposing the routine use of non-hierarchical ensemble techniques, rather than following the traditional single-solution pathways. It may have be of interest to other groups in the dendro-community and even in the wider field of proxy-paleoclimatology. I have only two minor comments both are related to the Supplementary.

We considered both suggestions.

line 973 The year should be corrected in the reference at the end of the line.

We corrected the reference: “(Barber 2012)”.

line1062: Please, explain what gamma means in the formula of the Hurst exponent.

We added: “(gamma is the power-law exponent that describes the decay of the autocorrelation function for long-term persistent data)”.

Reviewer #2 (Julien Emile-Geay)

Summary The paper investigates the impact of methodological choices on global temperature reconstructions based on tree-ring width proxies. It does so by enlisting a large consortium of authors implementing various approaches, and comparing the effect of these approaches on various metrics of global temperature change (extremes, amplitude, spectral properties). The topic is important, the study design is sound and the figures of high quality; yet, the manuscript itself falls flat in several respects. I recommend revisions to make the paper more broadly useful to the climate community at large, and these suggestions are numerous enough to qualify as major revisions.

We kindly acknowledge this comprehensive evaluation of our work. We carefully considered all comments and suggestions and have improved the manuscript accordingly.

After a careful read, I see three main issues that limit the current usefulness of the paper.

1.1 Dendrocentrism: Given the centrality of dendrochronology in high-resolution paleoclimatology, and given the limited traceability of many methodological decisions (e.g. detrending) in the published products, the approach taken here is very well conceived. However, I think the paper would be of broader interest if it were written for a broader audience. As it stands, too many assumptions are made about the reader's familiarity with the ins and outs of dendrochronology, particularly its jargon. To 99.9% of scientists, detrending means one thing; to dendrochronologists, it means removing the non-climatic component of growth – yet the reader would not know that from the current exposition. The same applies to a number of other jargon terms (e.g. “disc sample”). I recommend aiming the paper at a wider audience, not just the tiny fraction of the human population that speaks that lingo. I believe that is congruent with the goals of a non-disciplinary journal like Nature Communications. Surely there is a way to explain jargon in the methods section.

We cautiously revised the entire manuscript in order to avoid jargon or provide the necessary level of information that is needed for non-dendrochronologists to understand the full meaning (see marked version of the revised manuscript).

1.2 Weak conclusions: The paper's conclusions are rather weak, given the involvement of such a distinguished group of experts. As far as I can tell, the message is: methodological choices matter; one should use a similar approach (somewhat pompously called “non-hierarchical ensemble techniques”) to quantify their impact. It's hard to disagree with that. In fact, the real question is why it has taken the community so long to get to the point of truly collaborating in this manner. Yet it is excessive to claim that the paper “improve(s) our understanding of NH summer temperature variability before medieval times” (L229). What it does is shed light on the impact of the various choices, but beyond a dry description of which year is warmest/coldest in each reconstruction, there is (1) very little analysis of why the results end up being what they are and (2) what to do about it. That is, if a scientist outside the author group decided to re-use any of the reconstructions for a particular application, which should they use? If (as seems likely), there is no reason to prefer one reconstruction over another, how should they be optimally combined? One could use the whole plume, of course, but since the authors put forth the mean and median, presumably those are their favored options? If so, why? The arithmetic mean is optimal in the context of the central limit theorem, which assumes independent variations. Is that the case here? If not, what is the rationale for using the mean? Is there any sense in a weighted mean, as in Bayesian model averaging? It would behoove the paper to provide a rationale/interpretation for a reduced timeseries like the mean or median. In such a small ensemble, I worry that those reduced timeseries are effectively a referendum on which choice was made most often.

We show that the mean and median of the 15 reconstructions are effectively the same. With regards to “favoured options”, it is impossible to recommend any because there is no single quality on which to base a “favoured” judgement. One would face the same challenge in the Bayesian context when choosing suitable priors. If there were any credible non-TRW records of temperature variability for the past 1000 even 300 years, the job would be much easier, but

as there is not the best we can do is compare each reconstruction performance in terms high- and low-frequency signals to the instrumental data available. This limitation obviously precludes any serious discrimination based solely on a reconstruction's low-frequency performance, hence our reliance on the central limit theorem to identify extremes and give context to the mean and median. However, we added two new figures (Fig. 2 and Fig. S10), in which we highlight the reconstructions' similarities and dissimilarities in terms of the data used and methods applied. On this point we have amended the text significantly: "With the exception of R5 and R8, most ensemble reconstructions contain more long-term persistence than the observational evidence (see Fig. S9 for details of the reconstructions' individual power laws). Our community-driven experiment demonstrates how an investigator's decision-making process influences the behaviour and characteristics of climate reconstructions. Although an objective ranking of reconstructions is not possible since the true NH temperature history of the Common Era is unknown, we can make the following observations: Those reconstructions that correlate strongly with an independent instrumental target and exhibit relatively low AC1, such as the reconstruction mean and median, likely reflect better methodological choices in terms of signal preservation (Figs. 5, S10). Since R10 integrates instrumental temperature measurements during the calibration period, this reconstruction is not entirely independent of the target data. Note, too, that R9 and R10 likely both underestimate low-frequency variability due to the individual series detrending applied¹⁶, emphasizing that the selection of methods should not be based on calibration period statistics alone. In using five sites only, R7 evidently captures temperature variability across the smallest area of the NH extra-tropics. Those reconstructions that selected two-month or five-month seasonal windows are possibly under- or over-representing the JJA period that seems optimal for TRW formation at most of the nine sites. Last but not least, we believe that an AC1 value well above 0.7 and $H = 0.76$ are indicative of too much retained biological memory, e.g., R2, R6-7, R11-12, and R14-15 (Figs. 5, S10).", as well as "In conclusion, we advocate for the routine use of ensemble techniques and advanced data aggregation to develop more robust climate reconstructions. Based on our experiment, we consider the reconstruction mean as the most robust estimate and suggest using the upper and lower ranges of the 15 individual reconstructions to provide uncertainty estimates caused by the methodological choices of the investigators. Whenever possible, collaborative endeavours should focus on the development of new multi-millennia-long tree-ring chronologies (especially MXD) from those regions in both hemispheres presently under-sampled. In addition, we call for further development novel wood anatomical and isotopic records, and the combined assessment of different tree-ring parameters in advanced multivariate fusion approaches."

The paper also makes a lightning-fast comparison to the PAGES 2k reconstruction ensemble of Neukom et al. [2019]. I think many readers would be interested in a more in-depth comparison.

We added further description: "Our findings not only confirm existing evidence of summer temperature variability during the past millennium^{28,31,32}, but also provide new insights into the range of reconstruction variability before medieval times. Comparison with a multi-proxy product (i.e., PAGES19)¹⁵, which is based on various lower-resolution records and estimated global annual mean temperatures between 1 and 2000 CE (Fig. S11), reveals a reduced amplitude of evidence for large-scale cooling between the mid-13th century and around 1900

CE in the TRW-only reconstructions. Although this finding is consistent with previous observations of differing pre-industrial cooling trends between tree rings and lower-resolution temperature proxies²⁶, it is not clear whether this arises from methodological choices, the composition of the proxy network, or both. Despite a different temperature amplitude on decadal to centennial time scales in the TRW reconstructions compared with the re-scaled PAGES19 product (Fig. S11C), the strong positive correlation coefficients indicate agreement in the trajectory of Common Era temperature changes despite uncertainties in their magnitude. PAGES19 correlates with our reconstruction mean and median at 0.60 and 0.62 from 1-2000 CE, respectively. The relationship remains stable when using the first half of the Common Era ($r = 0.62$ and 0.58), but increases to 0.73 and 0.74 from 1001-2000 CE. Higher correlation coefficients are obtained after 50-year low-pass filtering the time-series (Fig. S11C). The PAGES19 reconstruction is characterised by an overall higher AC1 of 0.88 compared to 0.70 in the TRW-based reconstructions (1-2000 CE), which is likely due to the predominance of lower-resolution proxies in the first millennium of the Common Era and the global coverage of the PAGES19 reconstruction.”

Finally, there is throughout the paper an emphasis on the persistence properties of the reconstructions, whether measured by the lag-1 autocorrelation (called ‘AC1’ in the manuscript, though I don’t believe this notation is defined anywhere) or the Hurst exponent.

We added a further explanation of the Hurst exponent (H): “ $H > 0.5$ shows the presence of long-term memory in a system; see also methods below”, and also define AC1 as: “lag-1 autocorrelation (AC1)”.

What to make of these metrics? It is good to document them, but I did not see them used in a meaningful way. On L1324, it is said "Interestingly, on medium time scales between six and about 40 years the Hurst exponents are 1.42 (DFA2) and 1.44 (WT2)." Why is that interesting? *We added: “This is interesting because Hurst exponents as high as 1.42 or 1.44 are indicative of very strong persistence, and do not occur in any observational climate data, be it air or sea surface temperatures, precipitation totals, river run-off rates or sea ice extent. A Hurst exponent of 1.5 can be obtained if white noise data ($H = 0.5$) are summed.”.*

In summary, the paper goes to great lengths to document the methods and their impacts on various metrics, but falls short of making any meaningful scientific conclusions out of it. For instance, the response to volcanic cooling has been suggested as a way to probe transient climate sensitivity [Merlis et al., 2014]. Maybe this ensemble could be used for that purpose, quantifying the distribution of TCRs that results from the methodological uncertainty? That would be a tad more interesting than the current, dry list of statistics.

We now better describe the relevance of the various statistical parameters and also make specific recommendations for subsequent users of our ensemble reconstructions and their mean and median (see also previous responses). Furthermore, we added an entirely new Methods section and two new figures (Fig. 2 and Fig. S10). Please consider the marked version of the revised manuscript to see how we improved the article accordingly.

1.3 Reproducibility: The methods section is rather lengthy and repetitive. It is detailed enough to get a general idea of what was done, too detailed to be properly digested by a sane reader, and not detailed enough to reproduce the results. To do that, in 2020 (soon to be 2021), it is

absolutely essential that the authors share their code, preferably on a modern platform like GitHub.

Since the 15 groups used a wide range of different software programmes and instrumental target datasets when developing their reconstructions, it is not possible to make codes available. Please note that the tree-ring community is generally not programming in a specific language, but processes their data via the freely available ARSTAN software (<https://www.geog.cam.ac.uk/research/projects/dendrosoftware/>) before calibrating the resulting tree-ring chronologies against instrumental measurements. To ensure full transparency and reproducibility of the entire climate reconstruction process of each of the 15 groups, we provide detailed descriptions in the online supplementary material, and will also make all raw tree-ring width measurement series from each of the nine sites freely available at acceptance stage (<https://www.ncdc.noaa.gov/data-access/paleoclimatology-data/datasets>).

The description also makes it challenging to understand the causes of the differences between the various reconstructions, since several chunks of texts are common. Table S1 is a lot easier to parse, and maybe the methods section could be organized as a narrative explaining the various abbreviations in the table, and a brief rationale for each choice. That is, instead of focusing on what each reconstruction did, it would focus the choices available in each "column", and explain their relative merits/rationales.

We added an entirely new Methods section and designed two new figures (Fig. 2 and Fig. S10) concerned with highlighting the differences and commonalities amongst the 15 reconstructions. The individual descriptions of the 15 reconstructions have been moved to the online supplementary materials.

L93 "this technique has only rarely been considered for multi-proxy temperature reconstructions". I fail to see how Mann et al. [2008] (reference 5) exemplifies that; there are no ensembles in there.

*We replaced Mann et al. (2008) with "Werner, J.P., Divine, D.V., Ljungqvist, F.C., Nilsen, T., Francus, P., 2018. Spatio-temporal variability of Arctic summer temperatures over the past 2 millennia. *Clim. Past* 14, 527–557.", and also changed the reference list accordingly.*

L126 "A wide range of scaling and regression techniques was applied". Are they really that different in the grand scheme of things? Again, referring to Table S1, full of jargon abbreviations like "nested PCR", is assuming a lot of the audience. Having a method-centric methods section, as opposed to a group-centric one, would solve that.

We rewrote the sentence: "Different techniques of either scaling the TRW chronologies to the mean and variance of instrumental measurements, or regressing the TRW chronologies against the instrumental measurements were applied (see Fig. 2 and Table S1 for further details of the individual reconstruction methods)". If not done so in the main text, all abbreviations are defined in the newly added Methods section and the new figure 2.

L134-135 "their mean and median track the instrumental measurements remarkably well between the end of the 19th century and around 1990 CE": is that really a surprise, given that this is the calibration period?

We removed the word “remarkably”.

L135-136 “Proxy-target correlations are 0.76 and 0.79 for the reconstruction mean and median, respectively”. For series with this much autocorrelation, it is not out of this world. Please compare those numbers with Neukom et al. [2019], Table 1.

Done.

L150 “the ‘Divergence Problem’ did not exist prior to the Anthropocene”. Which Anthropocene are we talking about? The one that began in 1784 with James Watt’s invention of the steam engine, as proposed by Crutzen? Or the one that began in 1945 with the detonation of the Trinity bomb, which ushered the start of the Atomic Age? Or the Anthropocene that started thousands of years ago with the start of agriculture? You get the point: the start of the Anthropocene is a disputed notion. Why not use a calendar date, to lift ambiguity?

We changed the sentence: “Recent investigations suggest that methodology-induced challenges of proxy-target calibration, proxy network size, end-effects in time-series composition²², industrial pollution²³, or a combination thereof²⁰, likely explain this recent ‘Divergence Problem’.”.

L157 “spatial correlation fields of the ensemble mean spatial correlations with observed temperatures resemble the geographical extent obtained from a simple instrumental target-to-target comparison”. A tad abstruse. Suggest rewriting for clarity.

We rewrote the sentence: “Further evidence of the coherence between reconstructed and measured summer temperatures is the degree to which their spatial correlation fields correspond with an equivalent instrumental target-to-target correlation (Fig. 3D).”.

L174-176 “Interestingly, six different years between 1994 and 2016 CE were identified by nine individual reconstructions as the warmest summers, whereas eight and three reconstructions define 536 and 545 CE as the coldest summers, respectively.”. I do not find this trivia particularly interesting. What is the dynamical significance?

We removed the word “interestingly”.

L186 “This side experiment underlines the impact of different scaling periods, and provides further evidence of research bias that can override proxy information.” This is interesting. Do the authors have any explanation for this sensitivity? Any recommendations for the most appropriate scaling choice?

While our findings emphasize the influence of different scaling periods, and therefore stress the importance of transparency, an optimal period does not exist.

L196 “The TRW-based cooling signal, however, lasts least one decade, and appears robust”. Robust but demonstrably excessive, as the comparison to MXD-based reconstructions shows, only a few lines below. I fail to see the point of emphasizing this long-term signal, given what we know of the biological memory embedded in TRW (cf L209).

We deleted the sentence.

L209 this memory has major implications for comparison with simulated volcanic cooling, as shown by Stoffel et al. [2015] and more recently by Zhu et al. [2020]. Given the prominence given to this sort of SEA exercise in AR5, I think it's worth reminding the reader (again, most likely not a dendroclimatologist) that ring width is misleading at those scales, and that this is a known issue.

We added the following sentences: "If not adequately removed, the TRW-specific mid-frequency amplification due to biological memory⁷⁻⁹, can affect the interannual behaviour of climate reconstructions. One demonstrable example of this bias is the faster recovery of the MXD-based summer temperatures following the negative forcing of volcanic eruptions (Fig. S5). In line with the TRW data, the MXD data show their strongest cooling response to volcanic forcing in the first year after eruptions, but reveal substantially higher temperatures in the second year."

L215 "of diverse statistical properties" → "with diverse statistical properties", perhaps?

Deleted.

L226 "all ensemble reconstructions contain more long-term persistence than the observational evidence". Again, given the biological memory of TRW, is this a noteworthy result?

We rewrote the sentence: "With the exception of R5 and R8, most ensemble reconstructions contain more long-term persistence than the observational evidence (see Fig. S9 for details of the reconstructions' individual power laws)". Please also note that TRW chronologies can be pre-whitened to reduce their long-term persistence, i.e. the effect of biological memory can be removed.

L229 "Our findings [...] improve our understanding of NH summer temperature variability before medieval times". They really only improve our understanding of this particular set of reconstructions. More work needs to be done to explain NH summer temperature variability.

We changed the sentence: "Our findings not only confirm existing evidence of summer temperature variability during the past millennium^{28,31,32}, but also provide new insights into the range of reconstruction variability before medieval times."

L241-243 I wonder if the authors will venture to say something about the relative merits of TRW vs MXD reconstructions, and if a similar approach as used here is necessary for MXD-based reconstructions as well.

We added: "Whenever possible, collaborative endeavours should focus on the development of new multi-millennia-long tree-ring chronologies (especially MXD) from those regions in both hemispheres presently under-sampled. In addition, we call for further development novel wood anatomical and isotopic records, and the combined assessment of different tree-ring parameters in advanced multivariate fusion approaches."

Fig 1 the map requires an electronic microscope to consult. Suggest moving the map to an upper panel and squeezing the rest at the bottom. The rightmost panel makes it hard to judge how high the bars are by want of gridlines. A simple y grid aligned with major tick marks would solve that.

We re-designed the figure, which is now much clearer.

Fig 4F what is the list of volcanic events used here? It does appear rather consequential in such comparisons.

We added: "(see figure S3 for the 24 eruption years used)".

L414 and others "with the later normally containing" → with the latter normally ...

Corrected.

L1054 "The deviation of the Hurst exponent h from 0.5 (white noise) quantifies the strength of the long-term persistence of the record." this sentence is repeated verbatim in L1069. One will suffice.

Deleted.

Table S3 this is not easy to parse. Suggest using shades of blue/red to denote cold and warm extremes, so the eye can grasp them at once.

Done.

Fig S2 (right) the 1809 eruption is not highlighted, yet it clearly seems to be kickstarting the cooling.

Correct.

Fig S3 seems rather useless in view of the memory artifact illustrated in Fig S5.

Both figures provide different information: "Fig. S3. Superposed Epoch Analysis. (A) Average response of the 15 ensemble reconstructions (grey), as well as their mean and median (orange and red) to the 24 strongest volcanic eruptions of the Common Era. (B) Post-volcanic cooling relative to 10-year periods before all 24 eruptions (full), as well as after splitting into two early/late sub-periods of 12 eruptions each (</> 1170 CE). All eruptions exceed the Stratospheric Sulfur Injection (SSI) of 1991 Pinatubo event and occurred in 169, 266, 304, 433, 536, 574, 626, 682, 817, 939, 1108, 1171, 1191, 1230, 1257, 1286, 1345, 1458, 1600, 1640, 1695, 1815, 1835, and 1883 CE.", as well as "Fig. S5. Parameter-specific response. Average response of the MXD-based JJA large-scale NH reconstruction (green), as well as the 15 ensemble reconstructions (grey), and their mean and median (orange and red) to the 18 strongest volcanic eruptions back to 600 CE (expressed as anomalies with respect to the 10-year periods before the eruptions). All eruptions exceed the Stratospheric Sulfur Injection (SSI) of 1991 Pinatubo event and occurred in 626, 682, 817, 939, 1108, 1171, 1191, 1230, 1257, 1286, 1345, 1458, 1600, 1640, 1695, 1815, 1835, and 1883 CE."

Fig S6 the top number on the y axis should be 1.0, not 0.1.

Corrected.

Fig S7 The periodogram is a dreadful spectral estimator, being inconsistent. I wonder why the authors did not use a more reasonable one, like the Multi-Taper Method [Thomson, 1982], which might lump together some of these peaks. Also, the significance with respect to an appropriate null (AR(1), for instance) is not given.

We added a new figure (Fig. S8), in which the reconstructions' power spectra from the MTM are provided.

REVIEWER COMMENTS, second round -

Reviewer #1 (Remarks to the Author):

Dear Authors, Dear Editor,

I appreciate Authors' effort to improve the original submission during the revision stage. I think my concerns have been properly addressed in the revisions. I'm satisfied with the response or action taken. I think the revised manuscript is suitable for publication.

Reviewer #2 (Remarks to the Author):

[see next page/s]

Review of
**An ensemble approach to Common Era temperature
reconstructions from tree rings**
by Büntgen et al.

Recommendation: *Major revisions*

Summary This is a much improved version of the manuscript. I recommend publication after a few minor corrections/clarifications.

1 General Comments

Thank you for addressing my concerns so conscientiously. The manuscript is much improved and, from my perspective, is a landmark effort in the field. Too bad the editors of more upper-echelon Nature journals are too narrow-minded to give proper airtime to this type of work, as I believe it deserves a brighter spotlight. But I digress.

One comment concerns the characterization of the PAGES19 reconstruction effort as "based on various lower-resolution records". Note that PAGES19 is an ensemble effort very much like this one. Only 3 out of 7 methods used in the ensemble allow for low-resolution proxies, and in fact the largest LIA cooling in that ensemble is obtained with the BHM method of Barboza et al 2014, which used only annually-resolved observations. Thus, it is not accurate to characterize PAGES19 in this way; the picture is more nuanced.

I very much valued the comparison between this TRW-based effort and the MXD-based effort of Schneider et al 2015, particularly in regards to the recovery from volcanic cooling. However, there exist other MXD-based reconstructions from some of the present authors [*Stoffel et al., 2015*]. It may be helpful to explain why this particular reconstruction was chosen and not others (in the SI if necessary). I am also curious how the current effort stands in relationship to NTREND [*Wilson et al., 2016*], though this may be tangential to the manuscript itself.

Lastly, since this is, in part, an exercise in transparency and reproducibility, I reiterate my encouragement to archive code along with the data. NOAA Paleo places the bar for data archival relatively low, and divorcing the code from the data is one of the biggest mistakes our community continues to make time and time again. I realize this involves extra work, so I leave the authors free to address this; if complete archival is unrealistic, one compromise could be to archive a few of the 15 workflows. In my opinion, any would be far preferable than none.

2 Editorial Comments

L117-118 "There are ..." this sentence would actually be better suited as an opening to the section.

235-237 *While local monthly temperature measurements typically exhibit a Hurst exponent (H) of 0.55–0.75 during the 20 century, the NH summer mean contains more persistence ($H = 0.76$).* This phenomenon is well understood, and due to incoherent, short-term fluctuations averaging out in space [Fredriksen and Rypdal, 2016]. That it shows up in this context is therefore not surprising.

Fig S5 x-label says "relatie" instead of "relative"

Fig S8 caption mentions "95% confidence intervals", however they are point estimates (quantiles), not intervals. Also, from which test/null hypothesis are they derived? An AR(1) null, by the looks of it, but I cannot be sure.

L519 It is kind of the authors to include my name in the acknowledgments; however, I would prefer if it were correctly spelled out (Julien, not Julian).

JULIEN EMILE-GEAY

References

- Fredriksen, H.-B., and K. Rypdal (2016), Spectral characteristics of instrumental and climate model surface temperatures, *Journal of Climate*, 29(4), 1253–1268, doi:10.1175/JCLI-D-15-0457.1.
- Stoffel, M., et al. (2015), Estimates of volcanic-induced cooling in the northern hemisphere over the past 1,500 years, *Nature Geoscience*, 8, 784 EP, doi:10.1038/ngeo2526.
- Wilson, R., et al. (2016), Last millennium northern hemisphere summer temperatures from tree rings: Part i: The long term context, *Quaternary Science Reviews*, 134, 1–18, doi:https://doi.org/10.1016/j.quascirev.2015.12.005.

[Redacted]

Response to Reviewers comments, on behalf of all authors -
Professor Ulf Büntgen

Reviewer #1

I appreciate Authors' effort to improve the original submission during the revision stage. I think my concerns have been properly addressed in the revisions. I'm satisfied with the response or action taken. I think the revised manuscript is suitable for publication.

Many thanks.

Reviewer #2

Thank you for addressing my concerns so conscientiously. The manuscript is much improved and, from my perspective, is a landmark effort in the field. Too bad the editors of more upper-echelon Nature journals are too narrow-minded to give proper airtime to this type of work, as I believe it deserves a brighter spotlight. But I digress.

Many thanks.

One comment concerns the characterization of the PAGES19 reconstruction effort as "based on various lower-resolution records". Note that PAGES19 is an ensemble effort very much like this one. Only 3 out of 7 methods used in the ensemble allow for low-resolution proxies, and in fact the largest LIA cooling in that ensemble is obtained with the BHM method of Barboza et al 2014, which used only annually-resolved observations. Thus, it is not accurate to characterize PAGES19 in this way; the picture is more nuanced.

Changed to "Comparison with the PAGES19 multi-proxy product15, which reflects global annual mean temperatures between 1 and 2000 CE (Fig. S11)".

I very much valued the comparison between this TRW-based effort and the MXD-based effort of Schneider et al 2015, particularly in regards to the recovery from volcanic cooling. However, there exist other MXD-based reconstructions from some of the present authors [Stoffel et al., 2015]. It may be helpful to explain why this particular reconstruction was chosen and not others (in the SI if necessary).

We added: "Comparison of our ensemble reconstructions against the only reconstruction of extra-tropical NH summer temperature variability over the past 1400 years that exclusively uses MXD measurements²⁸."

I am also curious how the current effort stands in relationship to NTREND [Wilson et al., 2016], though this may be tangential to the manuscript itself.

See Büntgen et al. 2020 (Dendrochronologia) for a comparison.

Lastly, since this is, in part, an exercise in transparency and reproducibility, I reiterate my encouragement to archive code along with the data. NOAA Paleo places the bar for data archival relatively low, and divorcing the code from the data is one of the biggest mistakes our community continues to make time and time again. I realize this involves extra work, so I leave the authors free to address this; if complete archival is unrealistic, one compromise could be to archive a few of the 15 workflows. In my opinion, any would be far preferable than none.

As mentioned earlier, all raw tree-ring width measurements will be freely available via NOAA.

L117-118 "There are ..." this sentence would actually be better suited as an opening section.

We inserted a line break.

235-237 While local monthly temperature measurements typically exhibit a Hurst exponent (H) of 0.55–0.75 during the 20 century, the NH summer mean contains more persistence (H = 0.76). This phenomenon is well understood, and due to incoherent, short-term fluctuations averaging out in space [Fredriksen and Rypdal, 2016]. That it shows up in this context is therefore not surprising.

We agree and revised the sentence slightly: "Local monthly temperature measurements typically exhibit Hurst exponents (H) of 0.55–0.75 during the 20th century³⁰ (H >0.5 shows the presence of long-term memory in a system), whereas the NH summer mean contains more persistence (H = 0.76)."

Fig S5 x-label says "relatie" instead of "relative"

Corrected.

Fig S8 caption mentions "95% confidence intervals", however they are point estimates (quantiles), not intervals. Also, from which test/null hypothesis are they derived? An AR(1) null, by the looks of it, but I cannot be sure.

Corrected to "Dashed lines are the 95% significance levels relative to estimated (AR1) background noise."

UNIVERSITY OF
CAMBRIDGE

Department of Geography

L519 It is kind of the authors to include my name in the acknowledgments; however, I would prefer if it were correctly spelled out (Julien, not Julian).

Corrected.